# Quantitative, high-sensitivity measurement of liquid analytes using a smartphone compass

**Mark Ferris** [1,2] **& Gary Zabow** [1] ✉

Smartphone ubiquity has led to rapid developments in portable diagnostics. While successful, such platforms are predominantly optics-based, using the smartphone camera as the sensing interface. By contrast, magnetics-based modalities exploiting the smartphone compass (magnetometer) remain unexplored, despite inherent advantages in optically opaque, scattering or auto-fluorescing samples. Here we report smartphone analyte sensing utilizing the built-in magnetometer for signal transduction via analyte-responsive magnetic-hydrogel composites. As these hydrogels dilate in response to targeted stimuli, they displace attached magnetic material relative to the phone's magnetometer. Using a bilayer hydrogel geometry to amplify this motion allows for sensitive, optics-free, quantitative liquid-based analyte measurements that require neither any electronics nor power beyond that contained within the smartphone itself. We demonstrate this concept with glucose-specific and pH-responsive hydrogels, including glucose detection down to single-digit micromolar concentrations with potential for extension to nano-molar sensitivities. The platform is adaptable to numerous measurands, opening a path towards portable, inexpensive sensing of multiple analytes or biomarkers of interest.

The creation and uptake of portable, easy-to-use, analysis tools is democratizing science, enabling expensive, often inaccessible, laboratory-based testing to be performed with slashed costs, improved convenience, and with less required operator knowledge[1]. Among options such as paper-based diagnostics[2], wearable sensors[3], and other lab-on-a-chip devices[4], smartphone-based platforms are intriguing due to the ubiquity of phone ownership worldwide, including in remote and resource-limited areas. Their already in-built electronics and computational capabilities also offer the potential for more objective, quantitative measurements and comparisons than may be possible with other systems such as, for example, common paper-based test strips. Even so, expense associated with additional hardware requirements may still preclude their widespread deployment, such as when analyses are performed on separate equipment that simply use the phone's screen and processing capabilities as an interface, as for example with electrochemical glucose monitors[5], breathalyzers[6], fitness trackers[7], and genetic marker tests[8].

Truly integrated smartphone analyses, requiring no additional sensors, power, or electronics beyond those already embedded in the phone, may better enable wide dissemination. Significant progress has been made in this area, notably driven by Ozcan[9–13] and by Fletcher[14,15], with the creation of mobile microscopy tools that exploit the smartphone's built-in camera using simple attachments. Adoption of such camera-based optical techniques has now led to inexpensive smartphone-enabled test platforms[16] including mobile colorimetric[12,17] or spectroscopic[18], fluorescence[13,19], reflectance[20], surface plasmon resonance[21], chemiluminescence[22], and polarimetry[23] measurements. However, analogous use of built-in smartphone magnetometers beyond their intended compass usage[24,25] remains largely non-existent. Despite inherent advantages, including the ability to provide

[1]Applied Physics Division, National Institute of Standards and Technology, Boulder, CO 80305, USA. [2]Department of Physics, University of Colorado, Boulder, CO 80309, USA. ✉e-mail: gary.zabow@nist.gov

quantitative measurements directly from opaque samples with minimal or no prior sample preparation and being unaffected by autofluorescence and light scattering, the phone magnetometer has yet to be used for health or environmental analyte sensing or monitoring. Here, rather than using the camera or other potentially less quantitative optical modalities, we aim to bring magnetics-based sensing to the world of smartphone analysis by pairing the integrated phone compass with magnetized, shape-changing smart hydrogel composites.

Hydrogels are hydrophilic, three-dimensional, crosslinked polymer networks that swell in water while maintaining their connected structure[26]. With added functionalization, smart hydrogels can undergo reversible volume changes in targeted response to specific chemical or physical stimuli, making them desirable materials in multiple applications including, for example, as robotic grippers[27–29], and as actuators in sensor platforms[29–31]. While some smart hydrogels have impressive degrees of analyte-induced swelling[32], the response for other analyte targets, such as glucose, is often limited to more modest changes (~10% in length). However, this modest change can be amplified into larger motions by engineering variations into the hydrogel composition and/or geometry[29,30]. For example, as with metallic bimorphs within thermostats, an amplified curling motion can be created with bilayer hydrogel structures where two connected smart hydrogel layers swell to different degrees in response to a specific stimulus, or where a smart hydrogel is coupled to a second inert layer[33]. Indeed, the motion of such hydrogel structures can be described similarly to Timoshenko's original bilayer theory for metallic bimorphs[34]. Bilayer hydrogel structures have been created that respond to multiple stimuli, including humidity[35], light[36], pH[27], and temperature[27,37].

In limited instances, smart hydrogels have been combined with magnetic material to create sensors that transduce hydrogel dilations (expansions or contractions) into changing magnetic field signals[38–41]. However, these sensor constructs have thus far used only the dilation of uniform composition, homogeneous hydrogel materials, which shift the position of any associated (attached or embedded) magnetic material by relatively small distances, thus requiring strong magnets and/or sensitive magnetometers to detect the motion. By contrast, hydrogel bilayers can amplify the movement of any associated magnetic material, sometimes by orders of magnitude, similarly amplifying the change in any measured fields from those magnets. As shown here, this eliminates the need for large magnets, sensitive magnetometers, or even close proximity to a magnetometer and facilitates sensitive detection schemes with cheaper, less sensitive magnetometers, including those embedded within smartphones.

Point-of-use smartphone magnetometer-based sensing of liquid analytes is introduced here using a bilayer smart hydrogel structure (the hydrogel actuator, or analyte test strip) with embedded, magnetized particles that curls upon exposure to the target analyte, displacing the magnetized particles with respect to the phone magnetometer. The hydrogel actuator is demonstrated separately with both glucose- and pH-responsive hydrogels, but analogous structures can be created to target other analytes by changing the hydrogel chemistry. The dynamic range of the glucose hydrogel actuators cover typical millimolar physiological and pathological glucose ranges while also reaching high enough for bioprocessing monitoring (up to 50 mM). Due to the amplified motion of the bilayer design, the detection limit also extends down to the single-digit micromolar regime (a thousand times more precise than physiological glucose levels), and sensitivities should be further scalable well into the nanomolar regime. The material cost of the hydrogel actuator is on the order of a few cents and the device can be created without any complex or expensive processes. For widespread accessibility, the positioning attachments consist only of reusable, cheap 3D printable, or castable, plastic holders that require minimal user knowledge to assemble, being not unlike common phone protectors.

## Results

### Design of the magnetometer-based smartphone sensor platform

A prototype smartphone platform assembly, designed in this instance for the Motorola Moto E (2020), is depicted in Fig. 1A (see Supplementary Figs. 1 and 2 for smartphone platform assembly used for data collection). The full platform consists of a positioning clamp, the hydrogel actuator, a plastic attachment with a well for the analyte solution, and the smartphone itself. The clamp immobilizes the inert horizontal segment of the hydrogel actuator to the bottom of the analyte well in proper proximity to the magnetometer. Figure 1B shows a photograph of the prototype platform assembly. The hydrogel actuator (shown schematically in Fig. 1C) is formed in a flat T-shape geometry. It consists of an inert hydrogel on the horizontal segment of the T, used to clamp the actuator in place, and a hydrogel bilayer on the vertical segment, which comprises the actuating region. The bilayer segment consists of an inert hydrogel layer on the bottom and an analyte-responsive smart hydrogel layer on the top (show in gray and in yellow, respectively, in Fig. 1C). The bottm layer also includes an embedded, roughly disc-shaped ensemble of permanently magnetized, neodymium iron boron ($Nd_2Fe_{14}B$) microparticles, located near the bottom tip of the T-shape. These particles are silica-coated to prevent corrosion, minimizing any unintentional changes in their magnetic field properties when submerged in test solutions. The hydrogel actuator lays flat in the absence of analyte (Fig. 1C, left) and is positioned with the $Nd_2Fe_{14}B$ particles directly above the smartphone magnetometer. The bilayer section then curls upward when analyte is added (Fig. 1C, right), due to stimuli-induced shrinking of the top smart hydrogel layer (for smart hydrogels that expand in response to the analyte, the smart hydrogel layer would be positioned on the bottom). This curling moves the magnetic particles away from magnetometer, changing its magnetic field reading. Figure 1D shows a photograph of the hydrogel actuator before (left) and after (right) analyte activation. A video of the hydrogel actuator curling in response to analyte is shown in Supplementary Movie 1. Fig. 1E–I show screenshots from Supplementary Movie 1 of the actuator curling over time. An attachment piece was also designed for an alternative smartphone model, the Google Pixel 2, as shown in Supplementary Figs. 3 and 4.

For the proof-of-concept, we utilized a boronic acid-based formulation, which responds to glucose and other sugar molecules containing cis-diols, as the smart hydrogel layer in the hydrogel actuator. While the specificity is not the main goal of this paper, the hydrogel chemistry is based on the $GSH_{2.0}$ formulation devised by Nguyen et al.[38] which is optimized for selectivity over fructose and lactic acid. The inert portions of the hydrogel actuator were made from the same monomer components but without the sugar-binding boronic acid group. The amount of N-(3-dimethylaminopropyl acrylamide) (DMA) in the inert formulation was adjusted so that the initial post-cure pure water-driven swelling of the inert and analyte-responsive hydrogels match, resulting in a mostly flat hydrogel composite in the absence of analyte (see Supplementary Fig. 5a–e). Alternatively, the hydrogel composite can also be designed to be flat at any chosen glucose level (see discussion). We also demonstrate the versatility of the platform by reworking the chemistry to switch the selectivity of the actuator to pH. This pH hydrogel actuator was created from an acrylic acid-based smart hydrogel paired with an inert hydrogel layer such that it lays flat at pH 4 and swells with increasing pH. While the glucose hydrogel actuator has a shrinking glucose layer on top, here an expanding smart hydrogel layer is placed on the bottom of the hydrogel actuator so that the bilayer still curls upward and away from the phone surface (and magnetometer) as the pH increases. We achieved equivalent control over initial bilayer curvature of pH hydrogel actuators by tuning the relative cross-linker density between the inert layer and responsive layer.

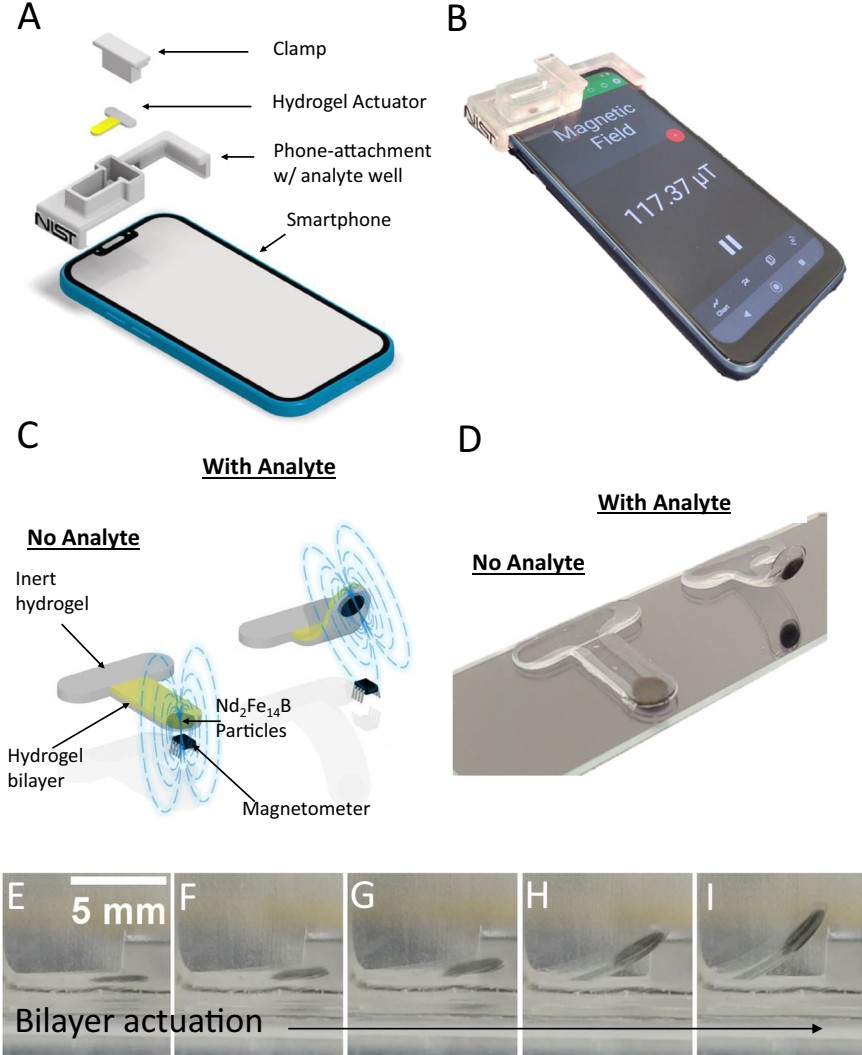

**Fig. 1 | Proof-of-concept design for magnetometer-based smartphone sensing.**
**A** Schematic of the full magnetic hydrogel smartphone sensor platform, consisting of an immobilizing clamp, the hydrogel actuator, a phone-attachment piece with a well to hold analyte solution, and a smartphone. **B** Photograph of a prototype sensor platform attached to phone. **C** Schematic of a T-shaped hydrogel actuator, with an inert region along the horizontal length and a bilayer region along the vertical length, consisting of a smart hydrogel *(top, yellow)* and an inert hydrogel *(bottom, gray)* with embedded $Nd_2Fe_{14}B$ particles. The bilayer region lays flat in the absence of analyte with the $Nd_2Fe_{14}B$ particles positioned directly over a magnetometer *(left)* and curls in the presence of analyte with the $Nd_2Fe_{14}B$ particles moved away from the magnetometer, decreasing its observed magnetic field.
**D** Photograph of the hydrogel actuator in the absence *(left, flat)* and presence *(right, curled)* of the analyte. **E−I** Curling of the hydrogel actuator over time, in response to analyte. For clarity, the images show a large degree of curling, though most of the useful signal change occurs for curlings represented in the first few panels.

## System characterization

Basic characterization of a glucose platform is shown in Fig. 2. Figure 2A shows changes in measured field strength ($\Delta B_z$) as the glucose concentration of a test solution is raised to 20 mM and then returned to zero (see Supplementary Figs. 6 and 7 for supporting data). Tested over three cycles, as shown in Fig. 2B (see Supplementary Fig. 8 for supporting information), the actuator accurately reproduces this response with standard deviation of the average endpoint magnetometer readings of no more than a half percent of the total response (see Fig. 2C). The hydrogel actuator used here had an average $\Delta B_z$ of 87 µT with a response time of roughly 20 min in response to 20 mM glucose, (see Supplementary Fig. 9 for supporting data). However, this observed change in field depends on how far the hydrogel actuator moves the magnetized particles as well as on the magnetic field generated by those particles, both of which can be increased many fold (see discussion below). Figure 2D shows that the glucose platform responds dynamically to changes in glucose concentration over several orders of magnitude. Similarly, basic characterization of the pH-responsive platform is shown in Supplementary Figs. 10–13. In addition to repeatability to within 1% of the pH-responsive platform over multiple trials on the Moto E, we also show similar repeatability on the Google Pixel 2 (see Supplementary Figs. 11 and 12), as well as high reproducibility, ~3%, between two smartphone models (Supplementary Figs. 12c–d). Finally, with the pH platform, we also demonstrate a ~2× speed improvement by simply adding polyethylene glycol (PEG) as a porogen to the hydrogel precursors, and a further ~2× speed improvement by making the bilayer thinner (see Supplementary Fig. 14a, b), while further improvements in response time are possible through optimization of these strategies (see discussion below). Alternatively, exploiting differences in initial response rates of the sensors, indicates that methods could also be developed for sub-minute tests (see Supplementary Figs. 14c, d).

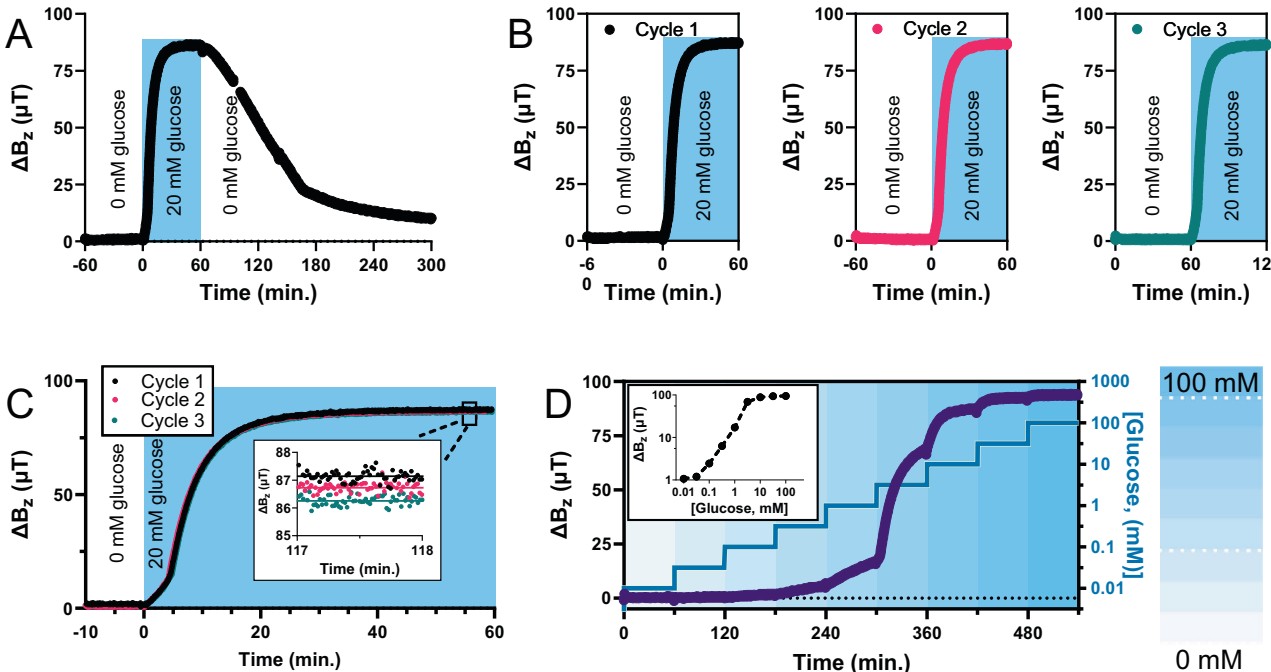

**Fig. 2 | Characterization of magnetometer-based glucose analysis. A** Actuation of the hydrogel bilayer in 20 mM glucose causes the hydrogel to curve upwards, changing the observed field strength ($\Delta B_z$) of the smartphone magnetometer, which then reverses upon returning to 0 mM glucose. The response is repeatable over three actuation cycles (**B**), with standard deviation of the average endpoint magnetometer reading of no more than a half percent of the total response (**C**). **D** Stepwise additions of glucose over time shows that the platform responds dynamically to glucose concentrations over several orders of magnitude. **D** (inset) Measured dose/response calibration curve (connecting dotted line is only to guide the eye). Source data are provided as a Source Data file.

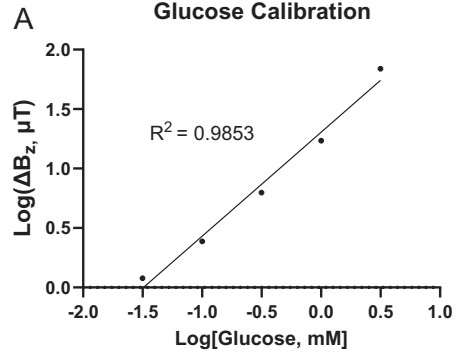

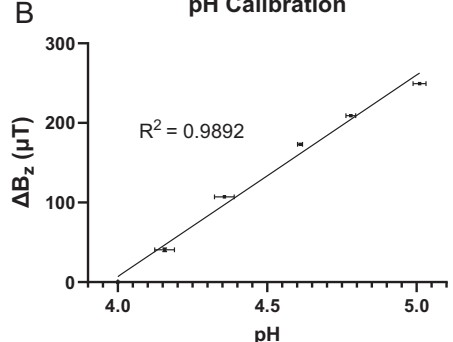

**Fig. 3 | Linear calibration of the glucose and pH sensor platforms. A** Calibration of the glucose sensor platform (data reproduced from Fig. 2D), showing the change in observed field strength $\Delta B_z$ to be approximately linear over two orders of magnitude of glucose concentrations. **B** The change in observed field strength $\Delta B_z$ of the pH platform is also approximately linear from pH 4 to pH 5. The pH test buffers used were measured in triplicate with a standard benchtop pH meter (standard deviation shown in the horizontal error bars) and plotted against the phone-based measurements, also performed in triplicate (vertical error bars). The negligible size of the vertical error bars on the scale of the plot, suggest that the phone-based measurements are more repeatable than the benchtop pH meter. Source data are provided as a Source Data file.

While, for clarity, Fig. 1D and Fig. 1G–I show a large degree of deflection from the bilayer in response to glucose, most of the $\Delta B_z$ induced by this iteration of the hydrogel actuator occurs with a smaller degree of deflection, in a range where the magnetic particles are moving mostly vertically with little change in their angle in relation to the phone surface and where the sensor response remains largely linear (see discussion). For example, the endpoint magnetometer readings of Fig. 2D (inset) are reproduced in Fig. 3A, with a straight-line fit revealing an approximately linear calibration ($R^2 = 0.985$) of the glucose platform over two orders of magnitude. Similarly, the pH platform yields approximately linear calibration ($R^2 = 0.989$, Fig. 3B and Supplementary Fig. 13) over a pH range that covers 80% of the sensor range.

**Detection limit**

Despite large dynamic ranges, the platform can detect remarkably low glucose concentrations due to the amplified motion of the bilayer design. As shown in the results from another hydrogel actuator (Fig. 4A, with supporting data in Supplementary Fig. 15), the detection limit for glucose reaches single-digit micromolar concentrations. This is already on par or better than most optical[42] and electrochemical glucose sensors[42,43] but can likely still be improved substantially. Detection sensitivity is limited by noise, which is here dominated by the inherent noise of the phone magnetometer rather than by the hydrogel actuators themselves, which can be seen to not appreciably change the overall noise when added to or removed from the phone (see Fig. 4B, with supporting data in Supplementary Fig. 16). Nor does

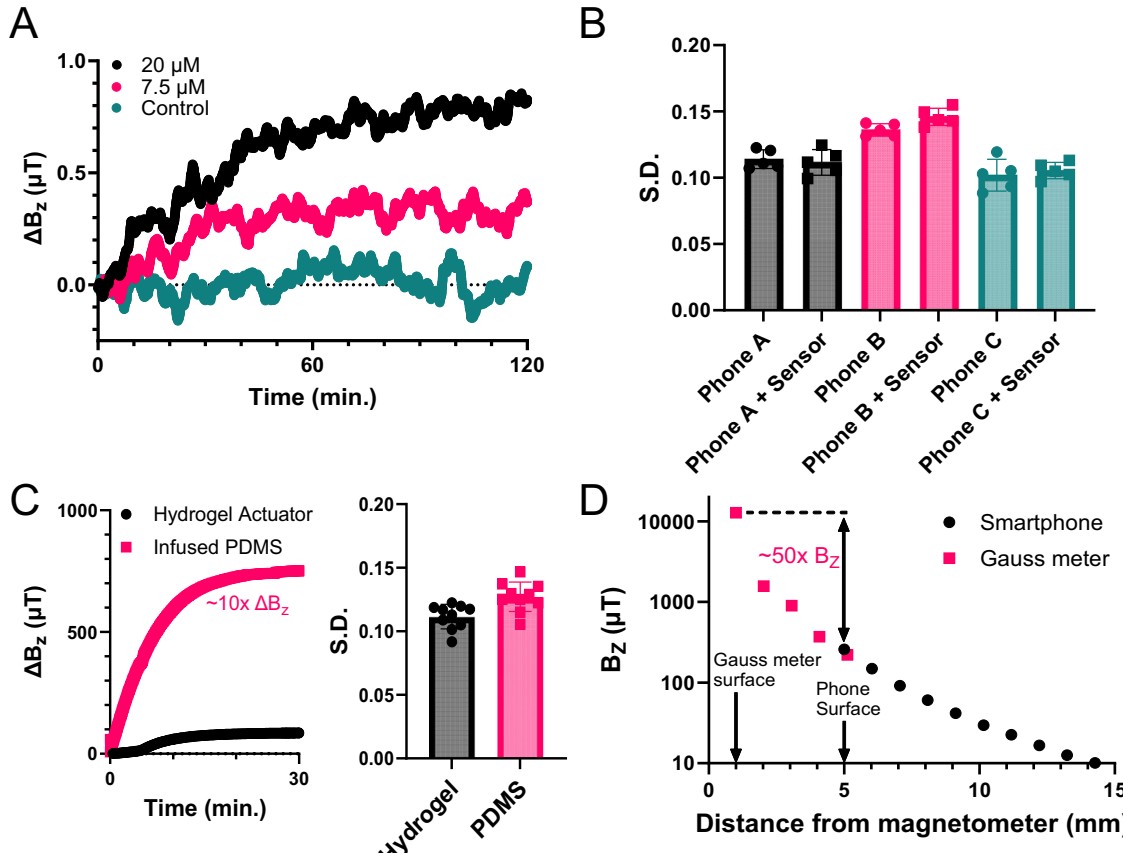

**Fig. 4 | Detection limit and noise analysis. A** Average change in smartphone magnetometer reading ($\Delta B_z$) over time for three replicates in response to different glucose concentration levels (labeled), showing that the platform can detect single-digit micromolar glucose concentrations. **B** Measurements of the system noise, determined by the standard deviation (S.D.) of the residuals in five different 100-s-long regions, for three different smartphones, each with and without a magnetic hydrogel actuator attached. The noise of the smartphone magnetometer does not increase appreciably when the actuator is added, indicating the noise is dominated by the embedded smartphone magnetometer. **C** (left) Increasing the loading of magnetic material in the hydrogel actuator by infusing the $Nd_2Fe_{14}B$ particles in PDMS and attaching externally increases $\Delta B_z$ ~10× (**C**, right) with only a minimal noise penalty. **D** Reducing the distance between the hydrogel actuator and magnetometer from ~5 mm closest approach with a phone to ~1 mm closest approach with a Gauss meter probe, yields a ~50× increase in magnetic field reading. Source data are provided as a Source Data file.

the noise change during actuation (see Supplementary Fig. 17). Therefore, with the noise dominated by the phone rather than the hydrogel actuator, numerous plausible routes exist to extend detection limits well into the nanomolar range through simple geometrical and/or magnetic changes to the actuator. Possible changes to the actuator geometry are detailed more fully in the "Discussion" section below but as first demonstrations, we include here two elementary magnetics-based modifications: (i) increasing the loading of magnetic material to increase the observed $\Delta B_z$ ~10× (Fig. 4C, left) with only a minimal penalty in terms of noise (Fig. 4C, right), and (ii) reducing the distance between the hydrogel actuator and magnetometer to yield a further two orders of magnitude increase in magnetic field reading (Fig. 4D).

### Real-world sample testing

As elementary real-world examples, in Fig. 5A, we demonstrate smartphone-enabled testing of sugar content in wine and champagne, without any pretreatment other than dilution in buffered water (because the particular actuator used was too sensitive). As expected, the high-sugar wine (sangria) induces a bigger $\Delta B_z$ than do the low-sugar options, pinot grigio and champagne brut. In Fig. 5B we demonstrate a simple first application of pH sensing with the smartphone-based testing of various common beverages, with no pretreatment. No buffers were used here since remnant pH buffer in

the well can alter the pH of subsequent test solutions; instead, $\Delta B_z$ was recorded from a baseline of tap water (pH~6.5) where the pH hydrogel actuator starts out curled and then flattens as the pH decreases. The pH's of the beverages were also measured with a standard pH electrode for comparison (see Fig. 5B). The lower pH beverages yielded larger changes in $\Delta B_z$, as expected.

### Discussion

Fundamentally, the sensor platform relies on the basic idea of a magnet being moved by a hydrogel. This in itself is not new[38–41]. A key to adapting such a system for smartphone-based sensing, however, and specifically for enabling high-sensitivity measurements with even a cheap magnetometer buried deep within a phone, is amplifying the change in the magnetic field by amplifying the motion of the magnet. For a block of hydrogel with uniform composition that dilates in response to some stimulus by some fraction, $\varepsilon$, of its length, $L$, an attached magnet would move a distance $\varepsilon \cdot L$. For a bilayer hydrogel strip such as those used here, however, a magnet attached to its end would instead move a vertical distance of approximately $3\varepsilon \cdot L^2/4h$ where $h$ is the thickness of the bilayer[34] (see Supplementary Information, Section 4.1). The magnet motion is therefore amplified over that achieved by attachment to a simple hydrogel by a factor of order $L/h$ or proportionally to the geometric aspect ratio of the bilayer strip. Increasing device sensitivity thus translates into a problem of

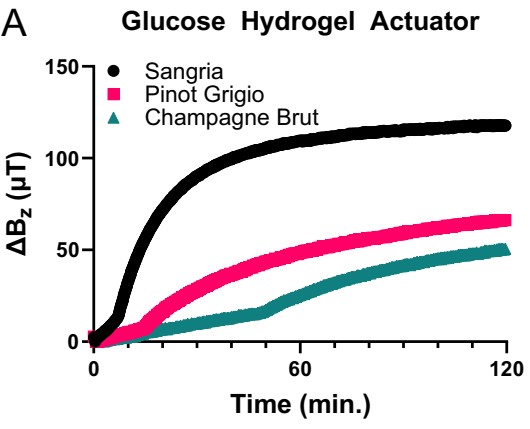

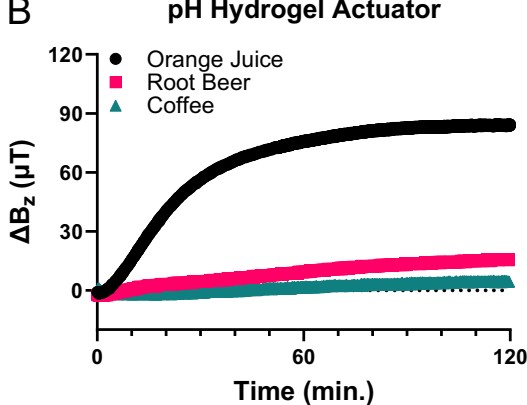

**Fig. 5 | Smartphone magnetometer analysis of common beverages. A** The response of the glucose platform to various types of wine confirms a higher glucose concentration in the high-sugar wine (sangria) than in the dry wines (pinot grigio and champagne brut). The inflection point is due only to an artifact in our measurement setup (see SI). **B** pH testing of orange juice, root beer, and coffee (pH = 4.00, pH = 4.42, and pH = 4.96, respectively, as measured by a pH meter). $\Delta B_z$ for each is recorded from a baseline of tap water (pH-6.5), with the lower pH beverages thus inducing larger $\Delta B_z$. Source data are provided as a Source Data file.

geometric design, rather than one fundamentally limited by intrinsic chemistry or material properties. This enables measurement sensitivities that are already, even with relatively short bilayer strips, of order 100- to 10,000-fold greater than comparable alternatives as well as measurement with different hydrogels that might offer only minimal material expansions (see Supplementary Information, Section 4.3).

In the proof-of-concept hydrogel actuator demonstrated above, the embedded magnet comprised a thin, disc-shaped pod of $Nd_2Fe_{14}B$ microparticles. This disk of microparticles was permanently magnetized out-of-plane, normal to the disk's flat surface, such that the disk's magnetic field points towards the magnetometer when the hydrogel is laid flat on top of the phone. The field from this system can be modeled by an imaginary Amperian surface current loop flowing around the disk circumference with a current density equal to the disk magnetization, $M$. The field $B_z$ at a distance $z$ away from the disk of thickness $d$ (assumed small) along its central axis is thus equal to that of a circular loop carrying a current of magnitude $i = Md$, which yields:

$$B_z(z) = \frac{\mu_0 i}{2} \cdot \frac{R^2}{\left(z^2 + R^2\right)^{\frac{3}{2}}} = -\frac{\mu_0 Md}{2} \cdot \frac{R^2}{\left(z^2 + R^2\right)^{\frac{3}{2}}} \quad (1)$$

where $R$ is the radius of the disk and $\mu_0$ is the vacuum permeability. Taking $Z_O$ to be the distance from the magnetic disk to the magnetometer within the phone, for small hydrogel fractional dilations, $\varepsilon$, and correspondingly small changes in distance, $\Delta z = 3\varepsilon\, L^2/4\, h$, the measurable change in field, $\Delta B_z \equiv (\partial B_z/\partial z) \cdot \Delta z$, is linear in $\varepsilon$ and given by:

$$\Delta B_z \approx \frac{9\mu_0 Md}{8} \cdot \frac{R^2 Z_0}{(R^2 + Z_0^2)^{\frac{5}{2}}} \cdot \frac{\varepsilon L^2}{h} \quad (2)$$

Differentiating with respect to $R$ yields an optimal disk radius, $R_{opt} = \sqrt{2/3} \cdot Z_0$, that maximizes this field change and yields a simplified expression linearly proportional to the original, analyte concentration-dependent hydrogel dilation, $\varepsilon$, with proportionality constant determined by the actuator's geometric and magnetic properties:

$$\Delta B_z \approx \frac{3^{7/2}\mu_0 MdL^2}{4 \cdot 5^{5/2} h Z_0^2} \cdot \varepsilon \quad (3)$$

Given the inverse squared distance dependence, reducing the separation $Z_0$ from the magnetometer significantly increases the measured signal, $\Delta B_z$. But, at least for smartphone-based sensing, this approach is limited by the depth to which the magnetometer is buried within the phone, which is around 3–5 mm for many phones. For the work presented here $Z_0$ was just short of 5 mm and thus using a phone model with magnetometer closer to its surface would improve sensitivity. Indeed, placing the actuator directly atop a bare magnetometer itself could increase $\Delta B_z$ by as much as two orders of magnitude (see Fig. 4D), but such setups are not readily available. Instead, restricting attention to smartphone-based sensing, amplifying the hydrogel motion provides a more accessible method to boosting sensitivity. Increasing hydrogel motion can, to some degree, be achieved through modifying the hydrogel chemistry (e.g increasing the amount of analyte-responsive monomer or decreasing the crosslinking degree) to increase $\varepsilon$, but this approach is materials limited and hydrogel- and analyte-specific. Instead, reducing the actuator thickness, $h$, or increasing its length, $L$, can boost sensitivity, linearly and quadratically, respectively, for any hydrogel material. Additionally, sensitivity can be increased by increasing the total magnetic moment, $m$, of the magnetic disk through increasing either the disk thickness, $d$, or its magnetization $M$, by increasing the volume fraction of magnetic microparticles comprising the disk (as seen in Fig. 4C, left).

The above assumes that the disk moves only vertically above the phone with negligible horizontal translation or angular reorientation of the disk. While true only for small displacements, such displacements do describe a considerable fraction of the sensor dynamic range. For example, as can be determined by substitution into Eq. 1, for the typical length-scales involved, a 50% drop in measured field is achieved by raising the magnetic disk just 1–2 mm above the phone surface. This entails minimal angular reorientation or horizontal translation of the magnetic disk. Also, provided the height that the disk is raised remains small compared to $Z_O$, the sensor remains close to linear in $\varepsilon$. Such linearity is experimentally evident in both the glucose and the pH sensing data (see Fig. 3A, B, respectively) and shown also theoretically in Supplementary Information, sections 4.1, 4.2, and Supplementary Fig. 20. For larger hydrogel bilayer curlings, however, there can be appreciable angular reorientation of the disk magnetic moment. Under such circumstances, Eq. 3 is no longer valid, and converting $\Delta B_z$ into a concentration value requires either prior calibration or a more complete mathematical treatment (see Supplementary Information, Section 4.1) that remains valid for arbitrarily large hydrogel curlings and accounts for both vertical and horizontal translation as well as angular reorientation.

Detection limits depend on the ratio of $\Delta B_z$ to background magnetic noise. Such noise may come from the phone magnetometer (and the phone itself), the magnetic hydrogel actuator, as well as random

background environmental sources, although such sources can often be rendered negligible by moving to a magnetically quieter spot in the room, without needing any magnetic shield. Once that is done, the data suggest that the noise is dominated by the intrinsic noise of the smartphone magnetometer, since random variations in the magnetometer readings do not increase when the magnetic hydrogel actuator is placed on the smartphone (Fig. 4C, with supporting data in Supplementary Fig. 16). Nor does the noise level depend on the height of the hydrogel magnet above the phone or the degree of curling in the bilayer (see Supplementary Fig. 17). This conclusion is further corroborated by the standard deviation of the residuals in $\Delta B_z$ reproducibly being on the order of 0.1 μT (Fig. 4B), which is the typical noise level for smartphone magnetometer chips[24]. Combined with the scalings in Eqs. 2 and 3, the negligible noise from the hydrogel actuators suggests that there remains considerable room to improve the system detection limits.

With its single-digit micromolar detection limit, the current proof-of-concept system is already of order 1000× more sensitive than typical blood glucose levels, but this was achieved with bilayers that were just 16 mm in length, $L$, and 0.65 mm thick, $h$. Without exceeding the size of a typical smartphone, actuator lengths could be increased at least 5-fold, while thicknesses could likely also be reduced by a similar factor, yielding a further combined sensitivity enhancement of order 100 based on geometrical considerations alone. Additionally, the disk-shaped magnets comprised only sparse ensembles of embedded magnetic particles none of which were necessarily high-grade neodymium, but simply what was readily available (see Supplementary Methods). These disks produced only small fields at the phone magnetometer of order 100 μT, whereas phone magnetometer chips typically measure up to as high as 5000 μT. This is not to say that field strengths could just be increased 50-fold without introducing additional magnetic noise, but initial tests are promising: attaching a polydimethylsiloxane (PDMS) disk infused with a higher volume density of magnetic microparticles to the actuator, which yielded 10 times the field at the magnetometer, increased the observed noise by only 10%–15% (see Fig. 3C), although this might also be due simply to insecure attachment of the disk to the hydrogel actuator compared to the truly embedded particles used above. Assuming the above geometric enhancements similarly do not introduce excessive additional noise, ultimate detection limits may reach well into the nanomolar range or beyond.

The dynamic range of the proof-of-concept glucose hydrogel actuator shown here already covers both normal and pathological ranges for both blood and saliva[44], as well as ranges typically encountered in bioprocess[31], but with maximum sensitivity around 0 mM, when the actuator is flat and as close to the magnetometer as possible. This actuator was designed to lay flat in the absence of glucose since highest sensitivity is typically desired for lower concentration range applications. However, the bilayer can be biased to lay flat, and therefore offer maximal sensitivity, at any desired glucose concentration by adjusting the DMA content in the inert layer (Supplementary Fig. 5a–e).

Hydrogel materials have widely tunable properties which should be explored in future work to improve the platform over the proof-of-concept described here. While the inherent portability and low skill required to use the platform makes it preferable to shipping test samples to a laboratory with specialized staff in terms of time required to complete an analysis, the response time should still be increased substantially for full impact. Supplementary Fig. 14 shows that adding PEG to introduce gel porosity already more than doubles the actuation speed, but optimization of this technique to the point where a continuous porous network is formed should improve the response time further, with a few minutes as an achievable goal[45]. In addition to increasing the motion of the hydrogel actuator (as per Eq. 3), reducing the bilayer thickness also reduces the distance for the analyte to diffuse into the hydrogel. Given the diffusive nature, response rates should increase quadratically with reductions in thickness, suggesting plausible response times on the seconds, rather than minutes, time-scales. Fabrication methods such as 3D-printing[46], or spin-coating[47], may help achieve thinner and more reproducible, structures. Thinner and/or more porous hydrogels may result in mechanically weaker structures, but this can be counteracted by increasing the strength of the hydrogel with established methods, such as by increasing the concentration of crosslinker, creating interpenetrating networks[48], and adding inorganic fillers[49]. Antifouling materials[50] might also be incorporated to better protect the hydrogels in complex, real-world samples.

Additionally, there remains much room to explore alternative actuator architectures and to add new analyte targets to the platform. While a bilayer curling motion was chosen here to amplify signal when compared to simpler uniform hydrogel slabs, alternative geometries would offer different features. For example, a buckling motion with a hydrogel pinned at two ends, instead of only at one, may enhance stability. Or, a twisting hydrogel motion may allow the field direction to switch from negative to positive, doubling the total possible signal change and increasing the dynamic range if the number of times the field direction switches between each 180° twist of the hydrogel actuator is tracked. In addition, in the current work, the field change is primarily along the z-axis of the magnetometer; simultaneous measurement along all three field axes, as is possible with most smartphone vector magnetometers, could enable multiplexed designs where multiple smart hydrogels that either actuate in different directions or contain magnetic particles magnetized in different directions are incorporated into a single device. Finally, in addition to glucose and pH, smart hydrogels are commonly designed to respond to other chemical and physical stimuli such as light[36], ionic strength[51], and temperature[27,37]. As with the glucose-responsive hydrogel, other analyte binding groups could also be incorporated to make gels responsive to other biologically relevant targets such as Na[+52], K[+52], lactate[53], and enzymes[54]. Smart hydrogels utilizing advanced targeting techniques, such as molecularly imprinted polymers[55] and complementary nucleic acid sequences[32], give the ability to tailor their response to specific proteins and DNA sequences, further multiplying possible use cases for this new form of smartphone-based sensing[1]. Potential applications are envisioned for home-based quantification of chemical contaminant levels in tap water; for food testing of other opaque liquids such as orange juice, milk, coffee, wine, or soups; for environmental analysis, such as for analyzing possibly murky lake or stream water in remote locations; and for mobile health analysis, such as with remote analysis of blood, sweat, urine, or saliva.

Ensuring accurate readings in such complex, real-world samples will require addressing issues of imperfect hydrogel selectivity, a consideration inherent to any hydrogel-based sensor. For example, boronic acid-based smart hydrogels also respond to ionic strength, temperature, pH, and other sugars, so these interferants should be co-monitored or the hydrogel chemistry modified to render their contributions negligible. Fortunately, steps are already being taken in this direction by, for example, the incorporation of quaternary amines into a boronic acid hydrogel[38,]. Alternatively, biological enzymes are known to be selective for biomolecules such as sugar and could increase sensor specificity. More generally, for the case of a phone sensor, signal and interferant contributions could also potentially be deconvolved by exploiting the vector nature of smartphone magnetometers to allow simultaneous measurement (as described above) of three hydrogels with different sensitivities to different interferants.

Finally, the design is simple and cheap: with all sensors, electronics, and power already included in the smartphone itself, it requires only the hydrogel actuator test strip and a positioning attachment, which can be recycled for repeat measurements. While new test strips may be desirable for each analysis to avoid cross-contamination, they

are inexpensive and reversible, making individual actuators also suitable for applications that require continuous monitoring. Estimated material costs of the hydrogel actuators are $0.16 for glucose hydrogel actuators (see Supplementary Table 1) and $0.03 for pH hydrogel actuators (see Supplementary Table 2), based on the price of the materials in the small quantities we purchased, though bulk purchasing would substantially further decrease costs. The concept presented here was demonstrated on two smartphone models, the Motorola Moto E (2020) and Google Pixel 2, though it is adaptable for any smartphone with a magnetometer, which is nearly all in the current day. The main consideration when adapting for a new phone model is ensuring that the magnetic particles are positioned over the magnetometer, which may be located differently on different phones, but the attachment piece can be 3D printed to fit any model. Additionally, the size of the liquid-containing well in the attachment, as well as the hydrogel actuators themselves, can be tailored to the desired samples.

The presented work brings magnetics to the world of smartphone-based sensing, introducing an alternative, optics-free, quantitative transduction mechanism that may enable numerous mobile sensing applications beyond just the glucose and pH sensing demonstrated here. Magnetics-based smartphone analysis allows for quantitative measurements within liquids that are opaque or otherwise optically unclear, often without the need for sample pre-processing and without concern for interference from variable background lighting, which may affect the accuracy of optically based measurements[1]. Using essentially just a smartphone compass, the initial proof-of-concept already shows dynamic ranges up to four orders of magnitude, and single-digit micromolar detection limits (for glucose). Potential scalability down to nanomolar detection ranges opens additional possibilities of working with highly diluted samples when original sample volumes might be small or hard to acquire. Combined with the ability to target hydrogels to different analytes and to potentially run multiple hydrogel actuators simultaneously, this raises the question of whether magnetics-based smartphone measurement might one day enable useful, home-based and/or point-of-use analyses with minimally invasive small samples such as minimal test quantities of fluid, or even individual drops of blood.

## Methods

### Materials
Acrylamide (AAm), N,N′-Methylenebisacrylamide (BIS), 3-(Acrylamido) phenylboronic acid (3-APB), N-[3-(Dimethylamino)propyl]methacrylamide (DMA), 2,2-Dimethoxy-2-phenylacetophenone (DMPA), Lithium phenyl-2,4,6-trimethylbenzoylphosphinate (LAP), 4-(2-Hydroxyethyl) piperazine-1-ethanesulfonic acid (HEPES), D+ Glucose, Tetraethyl orthosilicate (TEOS), acrylic acid (AA), Poly(ethylene glycol) diacrylate, average $M_n$ 700 (PEGDA700), and Dimethyl Sulfoxide (DMSO), and Sodium Hydroxide (NaOH) were purchased from Sigma Aldrich (St. Louis, MO). Sylgard 184 elastomer and curing agent (i.e. poly-dimethylsiloxane, PDMS) was purchased from Ellsworth Incorporated (Germantown, WI). Polyethylene glycol 10,000 (PEG), Thermo Scientific™ Orion™ pH Buffer (pH 4.01), and Thermo Scientific™ Orion™ pH Buffer (pH 7.00) were purchased from Thermo Fisher Scientific (Waltham, MA). $Nd_2Fe_{14}B$ microparticles with an average size of 5 µm were purchased from Magnequench (Ontario, CA), product number: MQFP-15-7. (These are not necessarily optimal grade neodymium, but simply what was in stock).

### Silica coating on neodymium iron boron ($Nd_2Fe_{14}B$) particles
For rustproofing, the $Nd_2Fe_{14}B$ particles were coated with $SiO_2$ through the hydrolysis and polycondensation of TEOS, also known as the Stöber method, by following a procedure described by Kim et al.[56]. Briefly, 40 g of $Nd_2Fe_{14}B$ particles are vigorously mixed in 1000 mL of ethanol with magnetic stirring. Next, 60 mL of 29% ammonium hydroxide is

slowly added, followed by 2 mL of TEOS. The mixture was then stirred for an additional 12 h before washing with ethanol.

### Glucose hydrogel precursor
The glucose hydrogel composition used in this study is derived from the $GSH_{2.0}$ hydrogel devised in Nguyen et al.[31]. Briefly, stock solutions of 2 wt% BIS in HEPES/NaOH, pH = 8.15 and 2 wt% LAP in deionized water were created for hydrogel syntheses. Hydrogel precursor solution was then made with the following: 172 mg AAm was dissolved in 500 µL of 2% BIS solution while 55 mg of 3-APB (for glucose responsiveness) was dissolved in 223 µL DMSO with 10 min. of sonication. These two solutions were combined before adding 95 µL DMA and finally 105 µL 2% LAP, used as a photoinitiator. The final mixture had a molar monomer composition of 73.3% AAm, 8.8% 3-APB, 15.9% DMA, and 2% BIS.

### Inert hydrogel precursor for glucose hydrogel actuator
The same stock solutions of 2 wt% BIS in HEPES/NaOH, pH = 8.15, and 2 wt% LAP in deionized water were created for hydrogel syntheses. 230 mg AAm was dissolved in 500 µL of 2% BIS solution, to which 223 µL DMSO, 52 µL DMA, and 105 µL 2% LAP (used as a photoinitiator) were added. The final mixture had a molar monomer composition of 90% AAm, 8% DMA, and 2% BIS.

### pH hydrogel precursor
0.4 g of AAm were dissolved in 0.9 mL DMSO, along with 10 mg of 2,2-Dimethoxy-2-phenylacetophenone (DMPA), used as a photoinitiator. 300 µL of PEGDA700 were added as a crosslinking agent, and 100 µL AA for pH-responsiveness. The final mixture had a molar monomer composition of 79.3% AAm, 0.2% PEGDA700, and 20.5% AA. For the introduction of PEG, 0.45 mL of DMSO was replaced with 0.45 mL of an aqueous solution of PEG at 0.4 g/mL.

### Inert hydrogel precursor for pH hydrogel actuator
0.4 g of AAm were dissolved in 0.9 mL DMSO, along with 10 mg of 2,2-Dimethoxy-2-phenylacetophenone (DMPA), used as a photoinitiator. 300 µL of PEGDA700 were added as a crosslinking agent. The final mixture had a molar monomer composition of 79.3% AAm, 0.2% PEGDA700, and 20.5% AA. The final mixture had a molar monomer composition of 99.8% AAm and 0.2% PEGDA700. For the introduction of PEG, 0.45 mL of DMSO was replaced with 0.45 mL of an aqueous solution of PEG at 0.4 g/mL.

### Synthesis of hydrogel actuators
Hydrogel actuators were synthesized in a T-shaped PDMS well with 0.5 mm depth. The bottom layer of the structures was partially cured under UV light (365 nm, 4 W) with $N_2$ gas flowing. Hydrogel precursor mixed with silica-coated $Nd_2Fe_{14}B$ particles was then deposited on top of the partially cured bottom layer, and particles were focused into a local disk with the placement of a separate permanent magnet placed below the PDMS mold. UV light (365 nm, 4 W) was applied with $N_2$ gas flowing to fully cure and bond both layers, entrapping the $Nd_2Fe_{14}B$ particles. The edges of the well were used to pin the hydrogel precursors for both layers. The $Nd_2Fe_{14}B$ particles were magnetized by briefly placing the hydrogel actuator into the bore of 3 T MRI scanner. The glucose hydrogel actuators were swelled in HEPES/NaOH, pH = 7.4 buffer solution after curing and conditioned by alternating between 0 mM and 20 mM glucose solutions 3×. pH hydrogel actuators were swelled in pH 4 buffer solution after curing and conditioned by actuating in pH 7 buffer and returning to baseline in pH 4 solution. See SI for detailed information on hydrogel actuator synthesis.

## Data availability
Source data are provided with this paper. Other data are available on request.

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

## Acknowledgements

M.S.F. acknowledges support provided by a National Research Council Research Associateship Programs Postdoctoral Fellowship and from NIST-PREP (Professional Research Experience Program), performed under the following financial assistance award 70NANB18H006 from U.S. Department of Commerce, National Institute of Standards and Technology. M.S.F and G.Z. acknowledge Internal NIST bioimaging funding. We thank Dr. Samuel Oberdick for his help with data processing and useful discussions. We thank Dr. Daniel Gruber and Dr. Montserrat Rivas for useful discussion. Any mention of commercial products does not imply recommendation or endorsement by NIST and are named solely for the purpose of fully describing the experiments.

## Author contributions

M.S.F. fabricated all experimental samples and phone fixtures, performed all experiments, and collected all data. M.S.F. and G.Z. analyzed the data and wrote the paper together. G.Z. conceived of the project and supervised the work.

## Competing interests

The authors, M.F. and G.Z., have filed a provisional patent application, serial number 63/529,254, through their employer, NIST, on aspects of this work.
