## [Peer Review File · Nature Communications]

REVIEWER COMMENTS

Reviewer #1 (Remarks to the Author):

In the submitted manuscript the authors present their results on combining several known techniques to create a smart phone-based analyte sensing system. These techniques include using smartphone sensors for analyte sensing, glucose and pH-sensitive hydrogels, bilayer structures to amplify the swelling response of the gels as well as transduction of the hydrogels' swelling response into electrical signals via magnetometers and embedded magnetic particles. The authors demonstrate that this combination can be used to sense pH and glucose with the help of a standard smartphone's magnetometer and thus provide a potential portable and low-cost analysis device.

The analysis of the performance of this setup is performed both with test solutions as well as with potential 'real-world' examples. Overall, the presented data is, especially taking the supplementary information into account suitable to demonstrate the functionality of the sensor as a glucose sensing device. Both the analysis of the data and its presentation is well performed. The literature referenced is appropriate for both the experiments and to convey the state-of-the-art.

The main challenges with this manuscript come from the fact that all the used techniques are – as mentioned above – already known and the novelty comes simply from the combination of those. While the authors properly disclose related work, this substantially lessens the impact and significance of this work and would require the demonstration of a substantial advantage of the combination over the already published results to warrant publication in a high impact factor journal.

In addition to this, the core experiments are sufficiently well described to reproduce, the manuscript is missing a few central descriptions to enable a complete experimental reproduction.

Therefore, I would suggest the following:

The main reason that the authors give for the necessity of a bilayer hydrogel cantilever system is that they claim that the standard volume change of the hydrogel would not be sufficiently detectable by the smartphone's magnetometer due to its location deep within the device. This leads to several comments/suggestions:

1) First of all, microfabricated magnetometry sensors are readily available and a stand-alone device with minimal distance to the magnetic material (as also demonstrated in this manuscript) could be easily fabricated and even connected to the smartphone, if its processing and data storage capacity is required. As the setup seems to depend on a rather large 3D printed frame there doesn't seem a large advantage between using an external magnetometer and the smartphone's internal one with this respect. This should be thoroughly discussed especially with respect to novelty, impact and significance of the work.

2) While the bilayer hydrogel definitely helps to increase the signal and thus the resolution and limit of detection of the setup, it is unclear if a simpler setup, such as the one presented in Ref 38 would not also be sufficient. This could be possibly combined with a larger or stronger magnet to create a

stronger signal and thus potentially even overcompensate the advantage of the cantilever. Here the authors should discuss this again with respect to novelty, impact and significance of the work.

3) Connected to the latter, while the bending of the bilayer cantilever is shown in the supplementary video a scale, or any discussion of the change in bending amplitude and angle is not discussed in the manuscript. This would help making a comparison to earlier work easier. Also adding images from the video to the manuscript could help to illustrate the sensors functionality.

Besides these concerns there are a collection further comments/suggestions:

4) It is quite obvious from the video that not only the distance but also the orientation of the magnetic material is changing during the bending of the cantilever. However, the theoretical discussion in equations 1-3 neglects this aspect completely. Also, the discussion is assuming a small change which seems not justified given the strong deflection. Here, the authors should improve this discussion to also include an angular dependence.

5) To help with reproducibility, the authors should a) mention the type of smart phone either in the materials section or early in the results section. Right now, it is only found in the conclusion section where it can be easily overlooked. b) at least the dimensions of the holding frame should be given or ideally a blueprint or design file to increase the reproducibility of the results

6) The use of a second smart phone to subtract the background as given in the supplement should be better explained and mentioned more prominently in the main manuscript as the required use of two smartphones again impacts the significance of the results.

7) While the 'real-world' examples illustrate the applicability of this concept, it would be interesting how this works with the mentioned physiological fluids (blood, saliva, urine).

8) To prevent a misinterpretation of the results by the readers, for the given real-world examples it should be discussed how cross sensitivities influence the results. For example, the glucose-sensitive hydrogel based on phenylboronic acid are also sensitive to salt concentration, pH and temperature. I would imagine that these are also different in the real-world samples and thus could lead to a wrong result. This should be made very clear.

To summarize, the manuscripts present an interesting idea for smart phone-based analyte sensing and demonstrates the feasibility of the approach with well performed experimental results and evaluations. However, that most of the individual concepts that have been combined to make this possible have been demonstrated and published before negatively impacts the significance of this manuscript. Furthermore, before I can recommend this manuscript for publication, the above mentioned issues should be addressed.

Reviewer #2 (Remarks to the Author):

The manuscript reports a smartphone analyte sensor platform based on magnetics, utilizing the built-in magnetometer for direct signal transduction via analyte-responsive magnetic-hydrogel composites. The integration of a bilayer hydrogel can amplify the motion of hydrogels dilate in response to their targeted stimuli, leading to the corresponding phone-detected changes in magnetic field. This capability enables the detection of subtle changes in hydrogel size, such as swelling or shrinking, without the need for optical equipment. It offers a sensitive, electronics-free method for quantitatively measuring analytes in liquid samples using only the smartphone's internal power, without any additional electronics or sample preparation. The smartphone-based platform is highly portable, making it suitable for on-site testing, thus has broad application prospects. However, the potential of the assay to be developed as POCT should be illustrated. Besides, some issues remain to be addressed.

Major issues

1. As about the accuracy, the precise calibration of the sensor platform is crucial to ensure reliable results. How can the calibration accuracy of this smartphone-based platform be ensured? In addition, please discuss variations in the magnetometer's performance across different smartphone models and manufacturers can lead to calibration issues. Robust calibration procedures are required to account for these variations and provide reliable measurements.
2. The manuscript demonstrated smartphone-enabled testing of sugar content in wine and champagne. However, real samples often contain various interfering substances, which can affect the performance of the sensor. Can this method be applied to other real samples? How to develop effective sample preparation techniques or sample handling strategies to minimize interference?
3. Ensuring that hydrogels respond specifically to the target and exploring durable and stable hydrogel formulations is critical. It is suggested to discuss more details about the specificity and durability of this hydrogel-based design.
4. Also about the robustness. The variability in hydrogel properties can affect reproducibility. How to ensure the reproducibility of hydrogel-based detection?

Minor issues

1. In line 97, the authors declared silica-coated particles can prevent corrosion and eliminate any unintentional changes in their magnetic field properties. Please provide further explanation of its principles and discuss whether other materials can be used to replace SiO₂?
2. Hydrogel responsiveness can be influenced by environmental conditions such as temperature and ionic strength. How does this design overcome this challenge?

3. In “Materials and Methods” section, the details of 3D printed attachment piece should be provided.

Reviewer #3 (Remarks to the Author):

The authors reports the demonstration of glucose and PH sensor using smartphone compass as optics free, no need of sample preparation, no further electronics and etc. It is a intriguing result using embedded magnetometer as a sensor in the smartphone because most of works using smartphone have took advantage of smartphone camera as detector or controller so far. I personally feel that the response time using magnetometer is too slow even though the authors claim that the speed can be faster using various methods such as changing the dimension of 'T'-shaped hydrogel actuator, designing hydrogel and etc. Even, to make the reliable change in vertical magnetic field, vertical displacement needs to be in a centimeter order. To make this work more valuable to publish in nature communications emphasizing the use of magnetometer in smartphone, technically advanced method for hydrogel actuator in material or architecture aspects needs to be suggested as a meaningful sensor in time scale.

The idea to use smartphone compass is still interesting that I recommend this to submit it in the more specialized journal.

We thank the reviewers for their time spent reviewing the manuscript and for their thoughtful, thorough comments. We also thank the editor for the opportunity to submit a revised version in response to these comments, which we feel have significantly improved the manuscript. Specific responses are detailed point-by-point below, but to summarize the main changes made in response to the reviewers, the updated manuscript now includes:

- A) An extensive mathematical analysis of the curling bilayer hydrogel motion, added to the Supplementary Information, that includes: (i) Derivation of analytic equations fully describing all changes in all vector magnetic field components measured by the magnetometer, valid for arbitrarily large bilayer curlings, displacements, and reorientations, (ii) An analysis of the range of validity of the original equations (1-3) in the original manuscript, and (iii) an exact mathematical comparison of the relative sensitivity of a bilayer versus monolayer hydrogel system.
- B) Additional experiments showing sensor operation also on a second phone model, indicating high degrees of sensor robustness, repeatability, and reproducibility across different phones.
- C) Addition of another sensor calibration curve together with a discussion of sensor linearity, showing approximately linear operation over much of the sensors' dynamic ranges.
- D) A new experimental demonstration of faster response time (added to the Supplementary Information), now bringing the sensors' T_{90} response time down to below 15 minutes.
- E) A discussion addressing issues of hydrogel targeting specificity and potential sensor interferants.

Here follow point-by-point responses interleaved between each reviewer comment (in bold).

Reviewer #1 (Remarks to the Author):

The main challenges with this manuscript come from the fact that all the used techniques are – as mentioned above – already known and the novelty comes simply from the combination of those. While the authors properly disclose related work, this substantially lessens the impact and significance of this work...

The reviewer is correct that certain concepts underlying our new sensor platform do not by themselves constitute new inventions. As we openly said ourselves in the original draft: *“Fundamentally, the sensor platform relies on the basic idea of a magnet being moved by a hydrogel. This in itself is not new [38-41]”*. What is new, however, is the first realization of magnetics-based smartphone sensing. While numerous optics-based systems exist that exploit the smartphone camera (many of which are reported in *Nature Communications* itself), no-one has ever attempted magnetics-based transduction through the phone compass, despite advantages that complement optics-based strategies. We argue that such compass-based sensing is not only a significant, novel development but one that is also timely and one that we hope will inspire many further applications not unlike those spawned by camera-based phone sensing.

... and would require the demonstration of a substantial advantage of the combination over the already published results to warrant publication in a high impact factor journal.

Being the first phone compass-based sensor there isn't any prior work to directly compare against, but we follow the reviewer's lead in comparing the work to Ref. 38 by Nguyen, Tathireddy, and Magda. Although that work is unrelated to cellphone-based sensing, it does involve a hydrogel-magnet combination, albeit a simpler monolayer hydrogel. As noted in our original draft, our curling bilayer system offers significant gain over any monolayer system. In our original draft, however, this gain was expressed in its most general form, in terms of dimensionless geometrical scalings, rather than specific

numbers, and it seems the absolute magnitude of that gain may thus have been missed. We have therefore now added explicit numbers together with an analysis to the supplementary information section showing that our system yields from 100 to over 10,000 times higher sensitivity than comparable alternatives. This, we believe, is not an insignificant difference and, even ignoring the cellphone sensing application, makes this paper more than just some “combination” of prior techniques. This analysis is also borne out experimentally with our reported glucose measurements approaching a thousand-fold more sensitive than any shown in Ref. 38, even though our results were acquired using a cheap magnetometer buried deep within a cellphone, rather than in direct contact with an external magnetometer as used in Ref 38. While this does not negate the value of a system such as that in Ref 38, which remains useful in other situations, it is these large gains that specifically make highly sensitive, portable phone-base measurement practical.

Further, we believe our use of shape-morphing bilayer hydrogels to amplify measurement sensitivity represents a key step-change in removing what have thus far always been fundamental hydrogel chemistry / materials limits on sensitivity; instead, our platform converts sensitivity gains into a problem of geometric design, agnostic to the underlying material. To strengthen these seemingly missed points we have now added the following to the manuscript immediately following our original discussion of geometric scaling factors: *“Increasing device sensitivity thus translates into a problem of geometric design, rather than one fundamentally limited by intrinsic chemistry or material properties. This enables measurement sensitivities that are already, even with relatively short bilayer strips, of order 100- to 10,000-fold greater than comparable alternatives as well as measurement with different hydrogels that might offer only minimal material expansions (see Supplementary Information, Section 4.3)”*.

In addition to this, the core experiments are sufficiently well described to reproduce, the manuscript is missing a few central descriptions to enable a complete experimental reproduction.

Therefore, I would suggest the following:

The main reason that the authors give for the necessity of a bilayer hydrogel cantilever system is that they claim that the standard volume change of the hydrogel would not be sufficiently detectable by the smartphone’s magnetometer due to its location deep within the device.

This leads to several comments/suggestions:

1)First of all, microfabricated magnetometry sensors are readily available and a stand-alone device with minimal distance to the magnetic material (as also demonstrated in this manuscript) could be easily fabricated and even connected to the smartphone, if its processing and data storage capacity is required. As the setup seems to depend on a rather large 3D printed frame there doesn’t seem a large advantage between using an external magnetometer and the smartphone’s internal one with this respect. This should be thoroughly discussed especially with respect to novelty, impact and significance of the work.

We thank the reviewer for these comments and the opportunity to clarify. There are two issues here:

First, we never actually claimed that standard hydrogel volume changes would not be detectable. What we said was: *“A key to adapting such a system for smartphone-based sensing, however, **and specifically for enabling high-sensitivity measurements** with even a cheap magnetometer buried deep within a phone, is amplifying the change in the magnetic field by amplifying the motion of the magnet.”* That is, a bilayer geometry significantly increases sensitivity, to the point of rendering cellphone-based measurement practical for high sensitivity measurements. Using a regular block of hydrogel on a

cellphone, by contrast, would require not only a far larger magnet, but also a larger hydrogel (so that the fractional change in that hydrogel's size translates to a large enough absolute motion to be detected). Not only is sensitivity much reduced, but with hydrogel responses being diffusion-limited, such larger hydrogels would also respond far more slowly, with response times increasing quadratically with hydrogel size.

Second, the reviewer is correct that it would also be possible to use a stand-alone magnetometer for measurement. And as the reviewer notes, this is something that we already did ourselves, with results already included in the paper for comparison purposes. But this misses a key point: we are specifically trying to create a simple, accessible, sensor system because, as we highlighted upfront in the paper introduction "*requiring no additional sensors, power, or electronics beyond those already embedded in the phone, may better enable wide dissemination*". As we mentioned above, we believe much of the novelty, impact and significance of the work lies precisely in enabling sensing through a cellphone alone, without requiring any additional electronics such that the system can be used as broadly as possible. This includes both in resource limited areas ---where the microfabricated magnetometry sensors that the reviewer refers to are unlikely to be readily available--- as well as in developed areas where the necessary know-how to connect any such standalone magnetometer electronics to the smartphone is a significant hurdle for many. We thus feel that it is incorrect to equate a simple plastic holder attachment with a complete external electronics unit (which itself would still additionally require its own plastic holder). Most everyone knows how to put a protective plastic screen cover on their phone (which is essentially equivalent to what we are proposing), and if need be many people could fashion a simple liquid well container to place on their phone themselves; only a tiny fraction of the general population, however, would know how to attach an external electronic magnetometer to their phone, assuming they could even obtain it. To reduce the chance of this distinction being missed by readers we have thus tried to further clarify by modifying the last line of the introduction to now read (original text in black, new text in red):

"The material cost of the hydrogel actuator is on the order of a few cents; and the device can be created without any complex or expensive processes,—and For widespread accessibility, the positioning attachments consist only of reusable, cheap 3D printable, or castable, plastic holders that require minimal user knowledge to assemble, being not unlike common screen protectors."

2)While the bilayer hydrogel definitely helps to increase the signal and thus the resolution and limit of detection of the setup, it is unclear if a simpler setup, such as the one presented in Ref 38 would not also be sufficient. This could be possibly combined with a larger or stronger magnet to create a stronger signal and thus potentially even overcompensate the advantage of the cantilever. Here the authors should discuss this again with respect to novelty, impact and significance of the work.

As mentioned in our response above, measurement is possible with a setup such as that in Ref 38, but for cellphones it would be orders of magnitude less sensitive as well as slower (see details in our response above). The suggestion of compensating with a larger magnet is, however, not a valid comparison. First, one could equally use a larger magnet with the bilayer strip to further increase its own sensitivity (as we did already demonstrate in the manuscript, Fig. 4c). Second, there is no way to compensate for the orders-of-magnitude relative gain of the bilayer system, let alone over-compensate for it, since the magnet needed to do so would be so much larger that it would saturate the range of the phone magnetometer many times over (typically phone magnetometers do not measure above 5000 μT). Even if the magnetometer were not saturated, the much larger magnet needed would require a larger hydrogel support, which would dramatically slow a monolayer system. Thus, again, we believe the orders-of-

magnitude sensitivity gain of our shape-morphing bilayer design is indeed of significant impact, being a key enabler for high sensitivity phone-based measurement.

3) Connected to the latter, while the bending of the bilayer cantilever is shown in the supplementary video a scale, or any discussion of the change in bending amplitude and angle is not discussed in the manuscript. This would help making a comparison to earlier work easier. Also adding images from the video to the manuscript could help to illustrate the sensors functionality.

We thank the reviewer for this good suggestion. We have now added images (together with scale bar) from the video to the manuscript, included now as sub-figures 1e-1i, and added a line to the text to refer to these sub-figures. In conjunction with this, to clarify the manuscript, we have moved the original sub-figures that demonstrated control of the initial bilayer curvature to Supplementary Fig. 5, and rearranged the subpanels and corresponding figure caption in Figure 1. We also rearranged the referencing text accordingly. Regarding discussion of bending amplitude/angles, please see comment (4) below.

Besides these concerns there are a collection further comments/suggestions:

4) It is quite obvious from the video that not only the distance but also the orientation of the magnetic material is changing during the bending of the cantilever. However, the theoretical discussion in equations 1-3 neglects this aspect completely. Also, the discussion is assuming a small change which seems not justified given the strong deflection. Here, the authors should improve this discussion to also include an angular dependence.

We thank the reviewer for raising this important issue and apologize for any confusion the video might have caused. The video was included to show hydrogel dynamics, specifically how a bilayer curves and the large displacement amplification that it enables. To simplify this for the reader, we chose a bilayer that underwent an easily visible bending, under which conditions the reviewer is correct that the orientation of the magnetic materials changes significantly. The confusion arises, however, from the fact that under normal operation the magnetic material displacement need not be that large (as explained below). To clarify the manuscript, we have now added the following text to the paper to the results section:

“While, for clarity, Fig. 1d and Figs. 1g-1i show a large degree of deflection from the bilayer in response to glucose, most of the ΔB_z induced by this iteration of the hydrogel actuator occurs with a smaller degree of deflection, in a range where the magnetic particles are moving mostly vertically with little change in their angle in relation to the phone surface and where the sensor response remains largely linear (see discussion).”

As now explained in the paper, as well as in a detailed mathematical analysis added to the paper SI, the field registered by the magnetometer is, for example, reduced to 50% of its initial value by having the bilayer hydrogel lift the magnetic material just 1 - 2 mm from the surface of the phone. For a typical bilayer this corresponds to only a small angular change and the majority of the sensor's dynamic field range thus corresponds to motion close to the phone where the displacement of the magnetic material remains essentially vertical. What we originally stated in the paper, however, was only that the included mathematical approximation was valid over small displacements, while regrettably failing to state that such displacements do cover much of the relevant range of the sensor. To correct this, we have now replaced existing text in the main paper following equations 1-3 to which the reviewer refers (see deletion in marked-up manuscript) with the following:

“The above assumes that the disk moves only vertically above the phone with negligible horizontal translation or angular reorientation of the disk. While true only for small displacements, such displacements do describe a considerable fraction of the sensor dynamic range. For example, as can be determined by substitution into Equation 1, for the typical length-scales involved, a 50% drop in measured field is achieved by raising the magnetic disk just 1 to 2 mm above the phone surface. This entails minimal angular reorientation or horizontal translation of the magnetic disk. Also, provided the height that the disk is raised remains small compared to Z_0 , the sensor remains close to linear in ε . Such linearity is experimentally evident in both the glucose and the pH sensing data (see Figs. 3a and 3b, respectively) and shown also theoretically in Supplementary Information, sections 4.1, 4.2, and Supplementary Fig. 20.”

The above also notes that the relatively small displacements give the sensor an approximately linear response. To better experimentally show this, we have thus switched the axes of the inset graph of figure 2d from its original log-linear to a log-log scale, which shows approximate linearity over two orders of magnitude of tested glucose concentration. We also include a new pH calibration curve showing similar linearity, which was included in response to a separate reviewer comment discussed below.

That said, the reviewer remains correct that for larger motions there is appreciable reorientation of the magnetic material and we have thus also added the following immediately after the above addition.

“For larger hydrogel bilayer curlings, however, there can be appreciable angular reorientation of the disk magnetic moment. Under such circumstances, Equation 3 is no longer valid, and converting ΔB_z into a concentration value requires either prior calibration or a more complete mathematical treatment (see Supplementary Information, Section 4.1) that remains valid for arbitrarily large hydrogel curlings and accounts for both vertical and horizontal translation as well as angular reorientation”.

As indicated in this text, for completeness we have now also added a full mathematical analysis to the supplementary information, deriving analytic equations for all vector components of the magnetic field that remain valid no matter how large the displacements or angular reorientations. In addition, the analysis added to the supplementary information also now explicitly discusses the range of validity of the original Equations 1-3.

We believe that these new math sections of the Supplementary Information, combined with the above text additions to the main body of the paper, has helped to clarify the sensor operation and remove any potential inconsistency for the reader and we thus thank the reviewer again for alerting us to this.

5) To help with reproducibility, the authors should a) mention the type of smart phone either in the materials section or early in the results section. Right now, it is only found in the conclusion section where it can be easily overlooked. B) at least the dimensions of the holding frame should be given or ideally a blueprint or design file to increase the reproducibility of the results

We thank the reviewer for these suggestions. We have now added a mention of the type of smartphone (Motorola Moto E (2020)) used for the glucose sensor characterization upfront in the first paragraph of the results section. We have also added a blueprint of the attachment piece/holding frame used for this data collection to increase the reproducibility of the results, as Supplementary Fig 2. Also, while the original manuscript already showed an equivalent attachment piece/holding frame for a Google Pixel 2 phone, which has similar overall dimensions but a different relative location of the magnetometer, we have now also added data generated using this alternative attachment piece and phone (see the response to reviewer 2, comment 1 below). To aid with the flow, we have moved the image of the Google Pixel 2 attachment piece to Supplementary Fig. 3 and added blueprints also for this attachment piece in

Supplementary Fig. 4. Accordingly, we updated one line immediately before the conclusion to reflection these changes (new text in red, original in black):

*“The attachment shown in this paper fits on a Motorola Moto E, though the design **The concept presented here was demonstrated on two smartphone models, the Motorola Moto E (2020) and Google Pixel 2, though it is adaptable for any smartphone with a magnetometer, which is nearly all in the current day. (see Supplementary Fig. 11 for an attachment piece designed for a Google Pixel 2, for example).**”*

In summary, now both smartphone models for which we adopted the platform are mentioned in paragraph 1 of the results, and images, dimensions, and blueprints of both attachment pieces can be found from there.

6) The use of a second smart phone to subtract the background as given in the supplement should be better explained and mentioned more prominently in the main manuscript as the required use of two smartphones again impacts the significance of the results.

We apologize for confusion over the use of two smartphones for data collection. The system certainly does not require the use of two phones, and this does not impact the significance of any results.

We originally used two smartphones for data collection to control for any potential fluctuations in the background magnetic field but found these fluctuations to be minimal over the course of development and testing of our device. If we were to show just the sensor phone signal (generated with just one phone) throughout the paper instead of the background subtracted signal (generated with two phones), the differences in the plots would be imperceptible and the conclusions would remain the identical. Please see Supplementary Figs. 2b and 2c (original manuscript)/Supplementary Figs. 6b and 6c (revised manuscript), comparing raw vs. background subtracted data, which are virtually identical and which we reproduce here for your convenience:

To better clarify this for the reader, we have modified wording in the Supplementary Information section detailing the background subtraction. Specifically, the section lead-in now reads (new text in red, original in black)

“To ascertain whether potential fluctuations in the background magnetic field might interfere with the sensor readings two smartphones were used for data collection...”

Similarly, the conclusion to that section now reads: *“While full background subtracted data is presented here for completeness, it should be noted that such background subtraction is only necessary in noisy magnetic environments. In practice there was never a substantial difference between the raw (sensor phone) and background-subtracted data in any of the data reported in this report, and therefore only one phone is necessary for implementation of the magnetometer-based smartphone sensor”.*

That said, if the editor prefers, we would be happy to replace any, or all, of the data figures with just the non-background-subtracted data acquired on a single phone.

7) While the ‘real-world’ examples illustrate the applicability of this concept, it would be interesting how this works with the mentioned physiological fluids (blood, saliva, urine).

We agree with the reviewer’s suggestion that it would be interesting to test the magnetometer smartphone sensor concept with physiological fluids, and ongoing work in our lab aims towards this goal. However, our main purpose for this manuscript was to first demonstrate, with a proof-of-concept, a new way of measuring liquid analyte concentrations via the smartphone magnetometer. And, as the reviewer states, we feel the already included ‘real-world’ examples are appropriate to show the applicability of the concept.

Regarding measuring the pH of physiological fluids, the pH hydrogel actuator needs to have a dynamic range that encompasses physiological pH (7.4), and thus we would need to rework the hydrogel chemistry with an acidic functional group with a pKa close to 7.4, which we believe to be more suitable to a follow-up study.

Regarding glucose measurement, as the reviewer correctly notes in comment 8 below, phenylboronic-acid-based hydrogels that are designed to be responsive to glucose also have cross-sensitivities to salt, pH, and temperature that will influence the results and will need to be accounted for in order to give an accurate determination of glucose concentrations. Please see our response immediately below for further comment.

8) To prevent a misinterpretation of the results by the readers, for the given real-world examples it should be discussed how cross sensitivities influence the results. For example, the glucose-sensitive hydrogel based on phenylboronic acid are also sensitive to salt concentration, pH and temperature. I would imagine that these are also different in the real-world samples and thus could lead to a wrong result. This should be made very clear.

The reviewer is correct that phenylboronic acid hydrogels are sensitive to salt concentration, pH, and temperature (as well as other sugar molecules). Accurate analyte readings in real-world samples, require such cross sensitivities to either be controlled for, eliminated through improvements to the hydrogel chemistry, or co-monitored so that their contribution to the observed signal can be mathematically removed. But such problems of hydrogel recognition specificity are universal to any hydrogel-based sensor, not only to this current work, and there is fortunately much ongoing work in the literature attacking precisely these problems through modifications to hydrogel chemistry / composition.

While we thus believe that fully solving such chemical specificity issues is beyond the scope of this paper, we do agree with the reviewer that we should have made these concerns clearer to the reader. Accordingly, we have now added the following text (which also includes a suggestion of how to use the vector nature of a phone magnetometer to control for interferants) immediately after our mention of real-world samples:

“Ensuring accurate readings in such complex, real-world samples, will require addressing issues of imperfect hydrogel selectivity, a consideration inherent to any hydrogel-based sensor. For example, boronic acid-based smart hydrogels also respond to ionic strength, temperature, pH, and other sugars, so these interferants should be co-monitored or the hydrogel chemistry modified to render their contributions negligible. Fortunately, steps are already being taken in this direction by, for example, the incorporation of quaternary amines into a boronic-acid hydrogel³⁸. Alternatively, biological enzymes are known to be selective for biomolecules such as sugar and could increase sensor specificity. More generally, for the case of a phone sensor, signal and interferant contributions could also potentially be deconvolved by exploiting the vector nature of smartphone magnetometers to allow simultaneous measurement (as described above) of three hydrogels with different sensitivities to different interferants.”

Reviewer #2 (Remarks to the Author):

The manuscript reports a smartphone analyte sensor platform based on magnetics, utilizing the built-in magnetometer for direct signal transduction via analyte-responsive magnetic-hydrogel composites. The integration of a bilayer hydrogel can amplify the motion of hydrogels dilate in response to their targeted stimuli, leading to the corresponding phone-detected changes in magnetic field. This capability enables the detection of subtle changes in hydrogel size, such as swelling or shrinking, without the need for optical equipment. It offers a sensitive, electronics-free method for quantitatively measuring analytes in liquid samples using only the smartphone's internal power, without any additional electronics or sample preparation. The smartphone-based platform is highly portable, making it suitable for on-site testing, thus has broad application prospects. However, the potential of the assay to be developed as POCT should be illustrated. Besides, some issues remain to be addressed.

We thank the reviewer for these complimentary comments on our manuscript and for the recognition of its broad application prospects. We believe that our answers to the reviewer's questions below of robustness, calibration, reproducibility, and environmental interferants, largely address the reviewer's question above regarding the system's potential for development for point-of-care testing (POCT).

Major issues

1. As about the accuracy, the precise calibration of the sensor platform is crucial to ensure reliable results. How can the calibration accuracy of this smartphone-based platform be ensured? In addition, please discuss variations in the magnetometer's performance across different smartphone models and manufacturers can lead to calibration issues. Robust calibration procedures are required to account for these variations and provide reliable measurements.

We thank the reviewer for raising these questions and have performed additional experiments to address calibration accuracy and variations in the magnetometer's performance across different smartphone

models. First, regarding calibration of the sensor platform, have added a new figure that shows a calibration of both the glucose and the pH sensor platforms, now labelled Fig. 3, with the original Figs. 3 and 4 now labelled as Figs. 4 and 5, respectively. Figure 3a shows a calibration of the glucose sensor with the data from the inset of Figure 2d, and with the updated log/log axes mentioned above. Figure 3b shows new data of the pH sensor platform calibration from pH 4 to 5 in pH increments of 0.2 pH units, with supporting data collection in Supplementary Fig. 13. This calibration experiment was performed in triplicate to show the reliability of the calibration. The differences in the trials were minimal, much smaller even than the difference in repeat measurements from a commercial benchtop laboratory pH meter, suggesting high robustness and repeatability of the hydrogel bilayer test strips and sensing platform. The data also shows that the response is approximately linear, simplifying the calibrations for both the pH and the glucose platforms over a large portion of the sensor range, as predicted by theory (see the response to Reviewer 1, point 4 and the additional mathematical analysis added to the Supplementary Information).

In tandem with adding this new Figure 3 showing the calibration of both pH and glucose sensor platforms, we have added the following text that can be found in paragraph 4 of the Results in the revised manuscript:

“For example, the endpoint magnetometer readings of Fig. 2d (inset) are reproduced in Fig. 3a, with a straight-line fit revealing an approximately linear calibration ($R^2 = 0.985$) of the glucose platform over two orders of magnitude. Similarly, the pH platform yields approximately linear calibration ($R^2 = 0.989$, Fig. 3B and Supplementary Fig. 13) over a pH range that covers 80% of the sensor range.”

We also tested the magnetometer’s performance across different smartphone models with a new set of experiments which have been added to the manuscript in Supplementary Figs. 11 and 12 together with an experimental description also now included in the revised Supporting Information, found below Supplementary Fig. 10. Additionally, we added the following sentence to paragraph 3 of the results of the revised manuscript to summarize and refer the reader to the more complete experimental description:

“In addition to repeatability to within 1% of the pH-responsive platform over multiple trials on the Moto E, we also show similar repeatability on the Google Pixel 2 (see Supplementary Figs. 11 and 12), as well as high reproducibility, ~3%, between two smartphone models (Supplementary Figs. 12c and 12d).”

To the reviewer’s point about magnetometers, ironically the hydrogel bilayer test strips provide measurements that are so repeatable and reproducible that they can even discern minute differences in the performance of the magnetometer chips buried inside the phones. As we discuss further in the SI, we did find a minor difference in the shape of the response curve between the two phone models, and minor differences (less than a couple percent) in the endpoint magnetometer reading. These results indicate that for the highest accuracy, a calibration should be made specific to each phone model. However, since the differences in magnetometer performance between the two models we tested were minimal, one could potentially use a single universal calibration for all smartphone models, since the small error penalty that would be incurred is likely tolerable for most real-world applications. To be sure, every phone model should be tested, but many different phone models use similar magnetometer chips, and thus differences found between other phone models are likely to be similarly small.

2. The manuscript demonstrated smartphone-enabled testing of sugar content in wine and

champagne. However, real samples often contain various interfering substances, which can affect the performance of the sensor. Can this method be applied to other real samples? How to develop effective sample preparation techniques or sample handling strategies to minimize interference?

The core concept demonstrated in this paper, that liquid analytes can be measured through the smartphone magnetometer, is not limited to only wine and champagne, and the method can certainly be applied to other real samples. These samples were arbitrarily chosen just to show an elementary example of smartphone magnetometer-based sensing, though measurement of other real samples is certainly possible (one other example of which is already included in the pH measurement of real-world samples). As the reviewer correctly states, however, real samples can contain various interfering substances, which can affect the performance of the sensor. We have strengthened the manuscript by adding a discussion of interfering substances in real samples as well as a suggestion on how to exploit the vector nature of the phone magnetometer to control for interferants (please see the response to Reviewer 1, Comment 8 for an explanation of the changes we made).

Regarding sample preparation techniques and sample handling strategies, these are dependent on the type of sample in question. However, an advantage of smartphone magnetometer-based sensing, as with magnetics-based sensing universally, is that it is immune to many types of optical interferences in complex real-world samples (i.e. optically opaque mediums, auto-fluorescing biomolecules, etc.) that can plague optics-based sensing. This renders much common pre-processing, often designed to remove optical interferants, unnecessary. Combined with inherent quantitiveness, this is a key advantage of magnetics-based analysis.

3. Ensuring that hydrogels respond specifically to the target and exploring durable and stable hydrogel formulations is critical. It is suggested to discuss more details about the specificity and durability of this hydrogel-based design.

We agree with the reviewer that the specificity, durability, and stability of the hydrogel response is critical for hydrogel-based sensors. We address specificity concerns in response to Reviewer 1, Comment 8. We believe the new data we added (described in response to Comment 1) in Supplementary Figs. 11 and 12 shows that our hydrogel actuator is also suitably stable and durable.

4. Also about the robustness. The variability in hydrogel properties can affect reproducibility. How to ensure the reproducibility of hydrogel-based detection?

As shown in the new data added as Supplementary Figs. 11 and 12, even though the hydrogel actuators are ultimately intended to be single-use, disposable tests, they are already quite robust. Specifically, the hydrogel actuator can be seen to accurately reproduce its response to the same analyte over multiple smartphones models and over at least 4 days while being repeatedly washed and resubmerged in test analytes solutions as well as repeatedly moved back and forth between different smartphones.

As the reviewer also states, variability in hydrogel properties between different hydrogel actuators would affect the reproducibility of the response between the different hydrogel actuators. However, we consider this to be a manufacturing challenge not specific to our device and outside the scope of this manuscript. For this first proof-of-principle demonstration, the hydrogel actuators were made by hand,

via a simple fabrication method that can be followed by the largest number of people. While such hand-fabrication is not the most reproducible, we see no reason why, in the future, laboratory automation, casting, 3D printing, or other methods could not be used to make many identical hydrogel actuators.

Minor issues

1. In line 97, the authors declared silica-coated particles can prevent corrosion and eliminate any unintentional changes in their magnetic field properties. Please provide further explanation of its principles and discuss whether other materials can be used to replace SiO₂?

The silica-coating method we used in the manuscript is a known process (the Stöber Method) and so we refer the reviewer to reference 56 in the revised manuscript for information about this method. Ultimately though, any coating method that prevents the particles from rusting could equally be used. As another example, particles can also be encased in polydimethylsiloxane (PDMS), an inexpensive, common electronics encapsulant, for rustproofing, such as we demonstrate in Fig. 4c of the revised manuscript.

2. Hydrogel responsiveness can be influenced by environmental conditions such as temperature and ionic strength. How does this design overcome this challenge?

We discuss hydrogel cross-sensitivity in our response to Reviewer 1, Comment 8 and have added a paragraph to the discussion that address hydrogel responsiveness in environmental conditions. Overcoming this challenge is relevant to any hydrogel-based sensor and is outside the scope of the manuscript, but with the additional paragraph in the discussion, we propose a method of overcoming the challenge by exploiting the 3-axis readout of smartphone magnetometers.

3. In “Materials and Methods” section, the details of 3D printed attachment piece should be provided.

We thank the reviewer for this suggestion. Please see the response to Reviewer 1, Comment 5, where we address the same concern. Blueprints for both the Motorola Moto E (2020) and Google Pixel 2 are now found in Supplementary Figs. 2 and 4 of the revised manuscript.

Reviewer #3 (Remarks to the Author):

The authors reports the demonstration of glucose and PH sensor using smartphone compass as optics free, no need of sample preparation, no further electronics and etc. It is a intriguing result using embedded magnetometer as a sensor in the smartphone because most of works using smartphone have took advantage of smartphone camera as detector or controller so far.

We appreciate that the Reviewer has found the use of a smartphone magnetometer, rather than smartphone camera as used in all other preceding work, to be intriguing.

I personally feel that the response time using magnetometer is too slow even though the authors claim that the speed can be faster using various methods such as changing the dimension of 'T'-shaped hydrogel actuator, designing hydrogel and etc.

We appreciate this concern about response time. Faster response times would always be better, and as discussed below we have now taken additional efforts to improve those times. But we respectfully disagree that the response time, even as it is now with initial prototypes, is too slow to be impactful.

In the original manuscript, we already showed in Supplementary Fig. 6 that adding polyethylene glycol as a porogen improved the response of the pH hydrogel actuator nearly two-fold. Now, we add new data showing that reducing the thickness of the bilayer improves the response time nearly two-fold again. Combined, this now yields a new T_{90} response time of roughly 13 minutes, a time already on-par with at-home COVID tests, which are a highly successful example of portable diagnostics, being used by billions of people around the world despite their 15-minute test time.

A difference between our system and a COVID test strip, however, is that our system provides a quantitative (rather than simply binary) measurement and that our current 13-minute test time is by no means the limit. As the reviewer notes, we did already mention in the original draft basic geometric scalings that can reduce this time further. Additionally, as discussed below, we also add now results indicating that, if the highest accuracy is not required, sub-minute test times are already possible.

Specific changes related to these speed increases now include an updated, original Supplementary Figure 6, which is now found as Supplementary Figs. 14a and 14b in the revised manuscript, with the added response curve from the thinner bilayer and now showing the T_{90} of all three responses. We also added two new panels to this Figure (Supplementary Figs. 14c and 14d), showing the pH hydrogel actuator response to different pH levels. This data shows that while the T_{90} of the sensor is on order ~ 15 minutes, a difference in the pH values can be determined in the first 30 seconds of the response by looking at the initial slope or ΔB_z at 30 seconds.

These updates are also reflected in the main text:

~~“The response time of the smart hydrogels can also be improved many fold by thinning the hydrogel layers or by increasing their porosity with established methods (as discussed below); as an example we demonstrate a 2x speed improvement by simply adding polyethylene glycol (PEG) to the hydrogel precursors (see Supplementary Fig. 6). Finally, with the pH platform, we also demonstrate a $\sim 2x$ speed improvement by simply adding polyethylene glycol (PEG) to the hydrogel precursors, and a further $\sim 2x$ speed improvement by making the bilayer thinner (see Supplementary Figs. 14a and 14b), while further improvements in response time are possible through optimization of these strategies (see discussion below). Alternatively, exploiting differences in initial response rates of the sensors, indicates that methods could also be developed for sub-minute tests (see Supplementary Figs. 14c and 14d).”~~

Similar text updates are also included in the supplementary information above Supplementary Fig. 14.

Even, to make the reliable change in vertical magnetic field, vertical displacement needs to be in a centimeter order.

We apologize for not including specific numbers related to vertical displacement in our original manuscript, an omission that seems to have caused some confusion among reviewers. As discussed in response to reviewer 1, comments 3 and 4, most of the sensor's range happens within a vertical displacement of order a millimeter, not a centimeter, making for a much more practical system. Additionally, reliable changes in the magnetic field can be observed with displacements much smaller than a millimeter (see Fig. 4a, revised manuscript).

To make this work more valuable to publish in nature communications emphasizing the use of magnetometer in smartphone, technically advanced method for hydrogel actuator in material or architecture aspects needs to be suggested as a meaningful sensor in time scale.

The idea to use smartphone compass is still interesting that I recommend this to submit it in the more specialized journal.

In our response to the Reviewer's comments above, we show new data bringing the T_{90} down to 13 minutes, on-par with at-home COVID antigen tests. We argue that this is a meaningful sensor in time scale, especially considering the goal is to replace laboratory testing, which is inaccessible to many, especially in resource-limited areas, and often takes many days to get results. Regardless, we agree that increasing the diffusion limited hydrogel response time is of high importance, and we will continue to improve the response time in follow-up reports.

We also agree that advanced methods and materials could be incorporated into the hydrogel actuator in the future to further improve its performance. We do note however that, as discussed above in response to reviewer 1 and as now included in the paper and in the supplementary information, our bilayer material design does already offer as much as 10,000 fold gain over simpler hydrogel constructions. This we regard as a not insignificant advance in hydrogel actuator material design. Additionally, there is a tradeoff between technically advanced methods and accessibility. Our goal was to present smartphone magnetometer-based analyte sensing as a high-sensitivity diagnostic tool with low-cost, and low technological barriers for both the creation and operation of the device and thus of all the possible advanced materials architecting options, we felt a bilayer system presented the best option. Finally, to the reviewer's point of journal choice, we do believe given the broad potential application, that this work is better suited for a more general, non-specialist audience, such as that of Nature Communications.

In addition to the above direct responses, for completeness in transparency, we note here a few additional remaining minor changes made to manuscript, purely to aid with the flow.

- i) We changed the name of the first subheader in the Results to "System characterization" to reflect changes already made to that section
- ii) We replaced an "order of" qualifier with its more exact numerical value. Specifically, we now say (original text black, new text red) "*For a bilayer hydrogel strip such as those used here, however, a magnet attached to its end would instead move a vertical distance of approximately $3\epsilon \cdot L^2 / 4h$ where h is the thickness of the bilayer³⁴ (see Supplementary Information, section 4.1)*"
- iii) We replaced an opaque proportionality constant, c , in the main paper equations 1-3, with its exact physical value, $\mu_0Md/2$, to enable more direct connection with the new mathematical analysis added to the Supplementary Information and modified the surrounding text accordingly.
- iv) We shortened the abstract to bring it closer to the desired 150 word limit.

Finally, we note that all changes made in the manuscript are indicated in the marked-up version that we are submitting alongside a new clean version, with all additions highlighted in yellow, all subtractions recorded in strikethrough font, and sections of text that were relocated verbatim marked with comments.

Once again, thank you to both the editor and all the reviewers for their time spent reviewing this work. Their comments are much appreciated and, we believe, have helped further strengthen the manuscript.

Sincerely

Mark Ferris, Gary Zabow

REVIEWERS' COMMENTS

Reviewer #1 (Remarks to the Author):

To whom it may concern,

I want to thank and congratulate the authors for successfully substantially improving their manuscript. Despite the well-made arguments by the authors in the rebuttal I still rate the novelty of the research as not incredibly high as both bilayer bending of hydrogel/other material combinations and magnetometer-based detection of hydrogels imbedded with magnetic elements is both known. Using the magnetometer in a smart phone instead of a stand-alone magnetometer is rather an obvious but a practical technical solution than a true innovation. However, this admittedly creates additional challenges that the authors successfully solved.

That being said, the improvements to the manuscript and the throughout clarifications as well as the combinations of the concepts to create a viable sensor solution has merit for publication. Furthermore, the sensitivity improvement due to the bilayer bending as clarified by the authors is impressive. This, in addition to the extensive mathematical analysis of the curling bilayer hydrogel motion, convinced me to recommend this for publication, despite the novelty issues.

Finally, I would like to suggest some minor changes to the added text:

Caption Figure 1: Considering what the authors wrote in their rebuttal, the bending in subfigures e) to i) seems to be not representative of what is actually happening in the sensor but rather an overemphasized illustration for demonstration of the bending. While this is mentioned in the text, it should also be noted in the figure caption to not mislead the readers.

With best regards

Reviewer #2 (Remarks to the Author):

Authors have addressed my issues about robustness, calibration, reproducibility, and environmental interferants about the sensors.

REVIEWERS' COMMENTS

Reviewer #1 (Remarks to the Author):

To whom it may concern,

I want to thank and congratulate the authors for successfully substantially improving their manuscript. Despite the well-made arguments by the authors in the rebuttal I still rate the novelty of the research as not incredibly high as both bilayer bending of hydrogel/other material combinations and magnetometer-based detection of hydrogels imbedded with magnetic elements is both known. Using the magnetometer in a smart phone instead of a stand-alone magnetometer is rather an obvious but a practical technical solution than a true innovation. However, this admittedly creates additional challenges that the authors successfully solved.

That being said, the improvements to the manuscript and the throughout clarifications as well as the combinations of the concepts to create a viable sensor solution has merit for publication. Furthermore, the sensitivity improvement due to the bilayer bending as clarified by the authors is impressive. This, in addition to the extensive mathematical analysis of the curling bilayer hydrogel motion, convinced me to recommend this for publication, despite the novelty issues.

Finally, I would like to suggest some minor changes to the added text:

Caption Figure 1: Considering what the authors wrote in their rebuttal, the bending in subfigures e) to i) seems to be not representative of what is actually happening in the sensor but rather an overemphasized illustration for demonstration of the bending. While this is mentioned in the text, it should also be noted in the figure caption to not mislead the readers.

With best regards

Response: We thank the reviewer for reviewing the manuscript and for their thoughtful comments, which we agree has led to an improved manuscript. In response to the reviewer's request, we have added the following line to the end of the caption for Figure 1:

For clarity, the images show a large degree of curling, though most of the useful signal change occurs for curlings represented in the first few panels.

Reviewer #2 (Remarks to the Author):

Authors have addressed my issues about robustness, calibration, reproducibility, and environmental interferants about the sensors.

Response: We thank the reviewer for reviewing the manuscript and are pleased that their issues about robustness, calibration, reproducibility, and environmental interferants have been address with our revisions.